# Species-Specific Humoral Immune Responses in Sheep and Goats upon Small Ruminant Lentivirus Infections Inversely Correlate with Protection against Virus Replication and Pathological Lesions

**DOI:** 10.3390/ijms22189824

**Published:** 2021-09-11

**Authors:** Rodolphe Michiels, Stefan Roels, Nick Vereecke, Elisabeth Mathijs, Laurent Mostin, Nick De Regge

**Affiliations:** 1Unit of Enzootic, Vector-Borne and Bee Diseases, Sciensano, Groeselenberg 99, 1180 Brussels, Belgium; rodolphe.michiels@ugent.be (R.M.); elisabeth.mathijs@sciensano.be (E.M.); 2Dierengezondheidszorg Vlaanderen (DGZ), Industrielaan 29, 8820 Torhout, Belgium; stefan.roels@dgz.be; 3Department of Virology, Parasitology and Immunology, Faculty of Veterinary Medicine, Ghent University, Salisburylaan 133, 9820 Merelbeke, Belgium; nick.vereecke@ugent.be; 4Experimental Center, Sciensano, Kerklaan 68, 1830 Machelen, Belgium; laurent.mostin@sciensano.be

**Keywords:** Maedi-Visna virus (MVV), Caprine arthritis encephalitis virus (CAEV), small ruminant lentivirus (SRLV), immune response, pepscan, cytokines, pathogenesis, transmission

## Abstract

Maedi-Visna-like genotype A strains and Caprine arthritis encephaltis-like genotype B strains are small ruminant lentiviruses (SRLV) which, for incompletely understood reasons, appear to be more virulent in sheep and goats, respectively. A 9-month in vivo infection experiment using Belgian genotype A and B SRLV strains showed that almost all homologous (genotype A in sheep; genotype B in goats) and heterologous (genotype A in goats; genotype B in sheep) intratracheal inoculations resulted in productive infection. No differences in viremia and time to seroconversion were observed between homologous and heterologous infections. Higher viral loads and more severe lesions in the mammary gland and lung were however detected at 9 months post homologous compared to heterologous infection which coincided with strongly increased IFN-γ mRNA expression levels upon homologous infection. Pepscan analysis revealed a strong antibody response against immune-dominant regions of the capsid and surface proteins upon homologous infection, which was absent after heterologous infection. These results inversely correlated with protection against virus replication in target organs and observed histopathological lesions, and thus require an in-depth evaluation of a potential role of antibody dependent enhancement in SRLV infection. Finally, no horizontal intra- and cross-species SRLV transmission to contact animals was detected.

## 1. Introduction

Maedi-Visna virus (MVV) and Caprine arthritis encephalitis virus (CAEV), also referred to as small ruminant lentiviruses (SRLV), are two related retroviruses infecting sheep and goats which are worldwide distributed but are predominantly found in developed countries. The main route of SRLV transmission is through the ingestion of contaminated milk/colostrum by newborns and less frequently through the inhalation of respiratory secretions. The airborne transmission route is usually favored in overcrowded stables where infected and healthy animals are raised together for a prolonged period of time [1,2,3,4].

These viruses are characterized by a slow and progressive infection inducing inflammatory lesions in the mammary gland, lung, joints and brain. After a long period of incubation, symptoms such as mastitis, pneumonia, arthritis and neurological disorders are commonly observed in one third of the affected animals [5,6]. While initially described as two distinct viral species of sheep and goats, respectively, MVV and CAEV were recently classified as prototype members of the group of small ruminant lentiviruses due to their ability to cross the interspecies barrier naturally and their genomic and structural similarities [4,7,8].

SRLVs are currently divided into five phylogenetic groups, from A to E, and are further subdivided into different subtypes based on the sequences of the *gag-pol* and *pol* genes [4,9]. Isolates from the genotype A group are now referred to as MVV-like strains, while isolates from the genotype B group are called CAEV-like strains. Several genotype A strains have also been detected in goats, and genotype B strains in sheep, confirming the potential of SRLV interspecies transmission [4,10,11,12].

Despite their close genetic relationships, it is known that the virulence varies between SRLV isolates [13,14]. In general, MVV (genotype A) is more pathogenic in sheep and leads to mastitis, pneumonia and neurological disorders. On the other hand, CAEV (genotype B) is considered more pathogenic in goats and results in clinical signs predominantly affecting the locomotive systems such as joints, the central nervous system and the mammary gland, while respiratory symptoms are less frequent. The limited amount of available data, mostly originating from field observations, indicates that heterologous SRLV infections (genotype B infections in sheep and genotype A infections in goat) mostly lead to subclinical infections. For example, subtype A4 strains were previously described as a low pathogenic in goats [14]. This is however not absolute since the presence of a pathogenic B2 strain was detected in sheep suffering from an arthritic outbreak [15]. Only limited information is currently available on the mechanisms underlying the differences in pathogenicity between strains, both after homologous (genotype A in sheep, genotype B in goats) and heterologous (genotype A in goats, genotype B in sheep) infections. Some hypotheses related to viral virulence factors were suggested to have an influence on the disease outcome. They include the presence of mutations, deletions or duplications in the LTR and *env* regions of SRLV that may affect the viral replication capacity and tissue tropism of these SRLV strains. Besides, the host genetic background and the compartmentalization of SRLV strains in target organs were also suggested to be determinant in the outcome of the infection [6,14,16].

Following a primary SRLV infection, both humoral and cell-mediated immune responses are activated to control the viral replication. The humoral immune response is first characterized by the production of non-neutralizing antibodies which mostly become detectable in serum about one month after infection [17,18]. These early antibodies are considered to be non-neutralizing since they are specific to linear epitopes localized in the capsid (CA), surface (SU) and the transmembrane (TM) proteins. The capsid protein p25 is considered as one of the most reactive antigens against which specific antibodies are rapidly produced and remain detectable for a long time [19]. The TM protein contains two immunodominant domains and appeared to be associated with arthritis in CAEV infected goats [19,20,21]. The SU protein includes five immunodominant domains that have been identified and consist of conserved and variable regions. Three highly variable regions, termed SU3 to SU5, have been shown to be highly immunogenic sites [19,21,22,23].

Besides the humoral immune response, a cell-mediated immune response is elicited to help to control the viral replication. Infected macrophages have been shown to produce cytokines which result in the activation of inflammatory responses. Previous studies have shown that SRLV infections induce a dysregulation in the expression levels of cytokines, causing the recruitment of additional inflammatory immune cells to the replication site and resulting in the infection of additional monocytes. The continuous viral replication associated with an increased inflammatory response causes the development of chronic lesions that gradually worsen over the years and lead to pathological changes in tissues [5,17,24,25].

In order to get more insight in the development of the immune response upon SRLV infection in sheep and goats, and the potential relationship thereof with differences in pathogenicity depending on the infected host, we performed a large-scale in vivo infection experiment lasting for 9 months whereby sheep and goats were infected with either a Belgian genotype A or genotype B SRLV strain. Besides the pathogenesis and immune responses, we also studied the efficiency of intra- and inter species horizontal SRLV transmission by introducing contact animals into the pens.

## 2. Results

### 2.1. Clinical Signs

During the whole experiment, animals were monitored for clinical signs such as lameness, respiratory distress and neurological symptoms referring to SRLV symptomatology. No clinical symptoms were observed in any of the animals over the 9 months of the experiment. Unfortunately, one goat and one sheep infected with the genotype B strain died at 2 and 32 weeks post inoculation *(p.i.)*, respectively. The autopsy revealed that the sheep probably died from a bacterial infection, and the goat died of a liver rupture after a fight with another goat.

### 2.2. Viremia and Seroconversion after Homologous and Heterologous Genotype A and B SRLV Infection

After inoculation with the genotype A strain, all six inoculated sheep seroconverted between 6 and 28 weeks *p.i.* (median: 11; mean: 13) and remained seropositive during the whole investigated period. For the goats inoculated with the genotype A strain, only one out of three seroconverted at 9 weeks *p.i.*, while the other two goats remained seronegative during the entire experiment (Table 1, Appendix A). This suggests that goats are less susceptible to infection by the used of genotype A strain than sheep.

After inoculation with the genotype B strain, all three inoculated sheep seroconverted between 4 and 11 weeks *p.i.* (median: 8; mean: 7.7), while the five inoculated goats became seropositive between 2 and 8 weeks *p.i.* (median: 4; mean: 4.6) (Table 1). Both sheep and goats thus seem equally susceptible to the genotype B strain, and no significant difference in time to seroconversion was observed between sheep and goats (Mann–Whitney test; *p*-value > 0.05). After the moment that seroconversion was first detected in each animal, they remained seropositive until the end of the experiment (Appendix A).

Moreover, qPCR analyses were performed to detect provirus and follow infections in all inoculated animals. All six sheep inoculated with the genotype A strain became qPCR positive between 2 and 24 weeks *p.i.* (median: 5; mean 8.2), and the only goat that was detected seropositive was also found qPCR positive at 4 weeks *p.i.* No provirus was detected in the blood of both goats that remained seronegative during the entire experiment (Table 1).

For animals inoculated with the genotype B strain, all three sheep were found qPCR positive between 4 and 5 weeks *p.i.* (median: 4; mean: 4.3), and all five goats became qPCR positive between 2 and 8 weeks *p.i.* (median: 5; mean: 4.8) (Table 1). Similar as in ELISA, the time to become qPCR positive did not significantly differ between sheep and goats infected with the genotype B strain (Mann–Whitney test; *p*-value > 0.05). After animals were detected positive for the first time in qPCR, some animals remained positive until the end, while others were intermittently tested positive (data not shown). All control animals remained negative in ELISA and qPCR during the whole experiment.

Altogether, these data showed that under experimental conditions, a SRLV infection is detected earlier by qPCR than by ELISA, and no clear difference in the time to detect seroconversion was observed between sheep and goats after homologous and heterologous infections.

### 2.3. Intra- and Cross-Species Horizontal SRLV Transmission

Results from ELISA and qPCR analyses on samples of contact animals that had been housed with the inoculated animals for 8 months showed that no SRLV transmission to contact animals was detected during the entire experiment (data not shown). This indicates that both intra-species and inter-species horizontal transmission was inefficient under experimental conditions.

### 2.4. SRLV Detection in Target Organs and Histopathology at 9 Months Post Infection

Since no differences were observed in the onset of viremia and seroconversion between productively infected sheep and goats with both genotype A and B strains, we subsequently evaluated whether strain specific differences in SRLV pathogenesis could be found upon infection of sheep and goats. We first performed qPCR analyses to detect the presence of virus genomes in four organs (lung, cerebellum, mammary gland and the synovial membrane) collected from the inoculated animals at 9 months post infection.

The genotype A strain was detected in almost all examined organs of infected sheep (22 out of 24 qPCR positive) but significant differences were present in the viral load found in different organs (Kruskal–Wallis test; *p*-value = 0.0047). The highest viral RNA loads were detected in the mammary gland with RNA loads equivalent to 10^2^–10^4^ TCID_50_/g, followed by the lung (10^1^–10^2^ TCID_50_/g, except sheep 90045 with a load equivalent to 10^4^ TCID_50_/g), cerebellum (10^1^–10^2^ TCID_50_/g) and synovial membranes (10^1^ TCID_50_/g). In contrast, significantly less (1 out 11; Fischer’s exact test; *p*-value < 0.0001) available organ samples of genotype A infected goats were found positive in qPCR. Viral RNA was only detected in very low amounts (equivalent to 10^0^ TCID_50_/g) in the synovial membrane of the goat that seroconverted after inoculation with the genotype A strain. The absence of viral detection in the two other goats was in line with their negative status in serology and the absence of viremia (Figure 1).

Similar results were found for animals infected with the genotype B strain. A total of 11 out of 20 organ samples of genotype B infected goats were qPCR positive. Three out of five goats were qPCR positive in multiple organs, while goats 08532 and 08539 were only positive in the mammary gland. As in genotype A infected sheep, the highest viral RNA loads after genotype B strain infection of goats were found in the mammary gland with values around 10^2^–10^4^ TCID_50_, although strong variation was found between animals (Figure 1). No statistical differences were detected in the viral load between the different organs (Kruskal–Wallis test; *p*-value > 0.05). After heterologous infection of sheep with the genotype B strain, no virus could be detected in any of the organs of the two remaining sheep (0 out 8), and thus observed significantly less positive organs than those found in the genotype B infected goats (11 out 20; Fischer’s exact test; *p*-value < 0.0097).

These results indicate that, despite viremia and seroconversion, heterologous infections with either genotype A and B strains result less frequently in virus replication and persistence in target organs.

To determine whether the presence of SRLV RNA in organs correlated with the development of lesions, a histopathological analysis was performed on lung, mammary gland and cerebellum samples.

In the mammary gland of genotype A infected sheep and several genotype B infected goats, apparent lesions were present and consisted of a massive lymphocyte infiltration with an atrophy and destruction of the acinar structures (Table 1; representative lesions are shown in Figure 2). In contrast, only minor lesions were found in mammary glands of genotype A infected goats and genotype B infected sheep. The severity of lesions observed in the histopathology of the mammary gland thus seemed to be in line with the observed viral load in the mammary glands of these animals.

Similar results were found in the lung of SRLV infected goats. Histopathological scores of genotype B infected goats were generally higher than those of genotype A infected goats and control goats. In sheep, this is more difficult to interpret. Although several genotype A infected sheep showed evidence of severe congestion with interstitial oedema and the presence of interstitial and peribronchiolar lymphocytic aggregates, also control sheep and genotype B infected sheep showed moderate lesions, making it difficult to estimate the additional impact of the SRLV infection.

No pathological lesions were observed in the cerebellum of any of the animals, not even in those found positive in qPCR.

Thus, despite the fact that no significant difference could be observed in viremia and the time to seroconversion, it seems that the genotype A and B strains were more capable to replicate and induce lesions in target organs (especially the mammary gland) after homologous compared to heterologous infection.

### 2.5. Cytokine Expression in Target Organs at 9 Months Post SRLV Infection of Sheep and Goats

To evaluate whether virus replication and virus induced lesions in target organs correlate with local cytokine production, we analyzed the mRNA expression patterns of interferons (IFN-α, IFN-β, IFN-γ) and inflammatory cytokines (TNF-α, IL-8, IL-10, TGF-β1, GM-CSF) in the mammary gland, lung and cerebellum of sheep and goats infected with the genotype A and B strains.

In the mammary gland of sheep and goats, a clear increase in IFN-γ expression could be observed after homologous genotype A and B strain infections. All six genotype A infected sheep showed a strong 10- to 30-fold raise in IFN-γ mRNA expression, while a 5- to 10-fold increase was observed in goats infected with the genotype B strain. No raise in IFN-γ expression was observed after heterologous infection as well as no clear change in expression for other tested cytokines could be observed, except for an approximately 5-fold increase in TNFα, IL-10 and TGF-β mRNA in some genotype B strain infected goats (Figure 3).

In the lung, no obvious changes in cytokine mRNA expression could be observed in most inoculated animals, neither after homologous nor heterologous infection with genotype A and B strains (Figure 4). Only one genotype A strain infected sheep (sheep 90045) showed a strong (>100-fold) raise in IFN-γ mRNA. Interestingly, this was also the sheep in which the highest viral RNA load was detected in lung tissue (10^4^ TCID_50_/g) (Figure 1). Furthermore, a limited 3- to 5-fold increase in IL-10 was observed in sheep infected with both genotypes.

Surprisingly, although only limited viral replication and no pathological lesions were found in the cerebellum, a strong 20- to 100-fold increase in IFN-γ mRNA expression was observed in four out six sheep inoculated with the genotype A strain. Similarly, also four out of five genotype B strains infected goats showed an increased IFN-γ mRNA expression, although to a lesser extent (10- to 50-fold). No increase in IFN-γ expression was found upon heterologous infection. Moreover, no increase was observed for the other cytokines that were studied (Figure 5).

In conclusion, these results show that an upregulated IFN-γ mRNA expression is the most important change in cytokine expression that can be detected at 9 months post SRLV infection. Such an increased IFN-γ mRNA expression is especially observed in lung and mammary glands of those animals in which the highest viral RNA loads were found after homologous SRLV infection.

### 2.6. Differential Humoral Immune Responses in Genotype A and B Infected Sheep and Goats

A commercial ELISA was used to determine the timing of seroconversion and to follow up the presence of SRLV-specific antibodies over time. Besides the ELISA, we also tested the sera in immunodiffusion (AGID) to follow the development of the humoral immune response. In general, the AGID results were in line with the ELISA results, although they detected seroconversion some weeks (mostly between 3 and 10 weeks, with one exception of 31 weeks) later than ELISA in most animals (Table 1). Interestingly, however, sheep inoculated with the genotype B strain were never found positive in AGID during the entire experiment, while they were found positive for the first time in ELISA between 4 and 11 weeks *p.i.*, and meanwhile all genotype B infected goats were found positive in both tests (Table 1). This provided an indication that the humoral immune response develops differently upon genotype B infection of sheep and goats. We therefore decided to study the antibody response against the CA and SU proteins after homologous and heterologous genotype A and B strain infections in more detail using the method of pepscan. Peptides corresponding specifically to the genotype A and B strains that were used as inoculum were used in this analysis.

First, sera collected at the end of the experiment, being 9 months *p.i.*, were tested. Overall, few reactive antibodies were found against the CA protein after genotype A infection in both sheep and goats. Only individual animals reacted strongly against some specific peptides, e.g., sheep 20540 against peptides 18 and 19 and sheep 90045 against peptide 45 (Figure 6). Alignment of the reactive peptides 18 and 19 with reference strains showed that these aligned with the known immunodominant region of the P25 capsid antigen of MVV strains (p28 in CAEV) (Table 2). In contrast, a more pronounced antibody response was observed against the SU protein after genotype A infection of sheep. All three sheep showed to have high amounts of antibodies against peptide 41, and 159–160. Moreover, some other peptides were found to react against the serum of two (e.g., 28, 132) or one sheep (40, 130, 131, 133, 134) (Figure 6). On the other hand, a genotype A strain infection of goats did not induce a consistent antibody response against the SU protein. Alignment of the reactive peptide sequences with amino sequences of reference strains showed that peptides 130–134 aligned with the variable region V4 of the surface protein, and peptides 158–160 were analogous to the conserved (C5) and variable region (V5) of SU5, the major immunodominant epitope detected in the surface portion of the envelope (Table 3).

Upon genotype B infection, sera of all three goats were reactive against peptide 18 of the CA; sera from two goats were reactive against peptides 17; one serum was reactive against peptide 19 (Figure 7). In sheep, no clear antibody response was raised against these epitopes. Mapping of this specific region using references’ strains showed that the sequence formed by the overlapping peptides 17, 18 and 19 shared the amino acid sequence NEEAERWRRNNPPPQAGGALTVD with the major immunodominant capsid antigen P28. This sequence differed on only two amino acids with strains from the B1, B2 and B3 subtypes (Table 2). In genotype B infected sheep, no consistent antibody response was induced against a specific region or peptide. Only some sera of individual sheep were found to react against some individual peptides (e.g., peptide 10, 11, 12, 23, 24, 25).

When looking at the SU protein, all sheep and goat sera seem to contain antibodies against the peptides P130 to P131. Peptides 129, 132 and 133 were also recognized as reactive but to a smaller proportion of animals. The OD values however revealed that more antibodies against these peptides were present in sera of genotype B infected goats compared to genotype B infected sheep (Figure 7). Alignment of the sequence formed by peptides 129 to 133 with reference genotype B strains showed that this region corresponded to the major immunodominant epitope SU5 in the surface portion of the envelope protein. This region of 25 amino acids was found to be more conserved in the first 13 residues, while the C’-terminal of the peptide showed more variations compared to the reference strains (Table 3). Besides the SU5 region, no other peptides were found to react consistently with multiple sheep or goat sera.

Taken together, these results for sera collected at 9 months post infection pointed out that stronger antibody responses against both CA and SU proteins were induced after homologous genotype A infection in sheep and genotype B infection in goats, than after heterologous genotype A and B infections.

Next, we analyzed how the antibody response had developed over time against several of the identified immunodominant epitopes in the CA and SU protein. Therefore, additional pepscan analyses were performed with sera collected at 1, 3 and 6 months (m) *p.i*.

For the genotype A infection (Figure 8) in sheep, only one sheep (20540) produced a specific antibody response against the capsid peptides 17–18–19, and it became only detectable at 9 months *p.i*. On the other hand, for different epitopes of the SU protein, an increase in the antibody response was observed for all three sheep starting at 6 months *p.i.* which further increased until 9 months *p.i*. For example, the antibody reactivity of sheep 90045 and 20540 against peptides 41, 130–134 and 159–160 roughly doubled every three months until the end of the experiment. For the genotype A infection in goats, the slight increase in antibody reactivity of goat 08521 against CA and SU immunodominant epitopes (Figure 8) was only observed at 9 months *p.i*.

A faster development of the humoral immune response was observed after infection with the genotype B strain (Figure 9). Upon infection of goats, one goat (08527) showed a high antibody response against capsid peptides 17–18–19 already at 1 and 3 months *p.i.* which decreased afterwards. Both other goats showed a modest increase starting from 6 months *p.i.* onwards. Moreover, specific antibodies against the SU5 region were already detected at 3 months *p.i.* for 2 goats (08527, 08532) and at 6 months for the third goat (08533). The response against this epitope quickly reached a high level that stayed stable until the end of the experiment at 9 months *p.i*. The antibody response against specific epitopes of the capsid and surface proteins upon genotype B infection in sheep was clearly more limited than in goats, but the limited amount of antibodies already became detectable during the first three months *p.i.* and remained at a low level afterwards.

## 3. Discussion

For years, MVV and CAEV have been considered to be two distinct viral pathogens infecting sheep and goats, respectively. Due to the increased use of sequencing tools and phylogenetic reconstruction methods over the last 25 years, molecular and epidemiological studies have however shown that MVV and CAEV are just two prototype members of a genetic continuum of viruses which have been named SRLV and are capable of overcoming the species barrier, resulting in interspecies transmissions [26]. Despite the known occurrence of interspecies transmission, the limited available data suggest a different outcome of infection depending on the strain and host that gets infected. MVV-like genotype A strains and CAEV-like genotype B strains seem more virulent in sheep and goats, respectively [13,15]. To increase our understanding of the mechanisms defining this strain and host specificity of SRLV induced pathogenesis and disease, we performed a unique large scale experimental infection study of sheep and goats with both a genotype A and B SRLV strain. Using the samples collected over the 9-month infection period, we studied differences in pathogenesis and immune responses in order to elucidate how the host species may influence the disease outcome induced by a specific SRLV strain.

Looking at the onset of infection, no significant difference in the time needed to detect viremia via qPCR or seroconversion via ELISA was found between homologous and heterologous infections with a same strain in those animals in which a productive infection was confirmed. The latter was the case for all goats (six) and all sheep (three) inoculated with the genotype B strains and all sheep (six) and one out of three goats inoculated with the genotype A strain. This seems to indicate that goats are less susceptible to infection with a genotype A strain when exposed to a low inoculation dose. It can however not be excluded that something went wrong during the intratracheal inoculation, but this seems unlikely since this method was previously described to be a highly efficient SRVL inoculation method for small ruminants. Furthermore, both goats in which no proof of viral infection could be found had coughed immediately after the inoculation, indicating the correct inoculation of the virus [27].

Despite the inoculation of a higher infectious dose of the genotype B strain (dose: 10^4.66^ TCID_50_/animal) compared to the genotype A strain (dose: 10^2.66^ TCID_50_/animal), no differences were observed in the Ct values found in leucocyte pellets during the experiment. Ct values remained close to the limit of detection, indicating that only a low number of monocytes/macrophages carrying provirus were present in the infected animals. This is in line with studies showing that only 1 out of every 10^4^–10^5^ PBMCs harbors provirus in SRLV infected animals [2,28]. These results indicate that if a productive infection of a host is initiated, no differences in basic aspects such as infection of monocytes/macrophages or seroconversion occur between homologous and heterologous infections. This could be seen as an advantage for SRLV detection in control programs since it shows that also heterologous infections should be detected with the same efficacy as homologous infections if the circulating strain can be detected by the used diagnostic tests. Furthermore, although qPCR was able to detect provirus somewhat earlier than the antibody response with ELISA, ELISA would be recommendable since antibodies remained detectable throughout the experiment, while this was not the case for provirus detection via qPCR [29]. For the sheep (20196) infected with the genotype A strain, we have no clear indication why it took longer before proof of a SRLV infection was found in this specific animal. The most plausible explanation is that fewer target cells became infected upon inoculation compared to other animals, meaning that the amount of proviral DNA remained for a long time under the limit of detection of the qPCR test. The virus that was present at undetectable levels in qPCR was probably however sufficient to elicit an antibody response that was detected by AGID and ELISA, but also somewhat later than in most other animals.

Once sheep and goats are infected, it is known that SRLV induce a slow progressive disease, affecting multiple organs such as joints, lungs or the mammary gland, which usually takes several years before clinical signs get visible [5,30]. In line with this, no clinical signs were observed during the 9 months of our experiment. To get insight in the potential clinical disease that could have developed over time if the experiment had lasted longer, histopathology was performed on samples collected at euthanasia. Severe lesions were observed in the mammary gland, both after homologous genotype A infection in sheep and genotype B infection in goats, indicative of the fact that the SRLV infection could have led to mastitis in the long term. The severe lesions observed in the mammary gland correlated with the fact that the highest viral RNA loads were found in this organ in qPCR. As shown in other studies, this most probably reflects virus replication in the epithelial cells of this target organ [24,31,32]. On the other hand, lesions in the mammary gland were absent or minor after heterologous infection, showing that despite the capacity of the SRLV strains to infect the monocytes/macrophages of their heterologous host, infection and/or replication in cells of target organs is restricted. This lack of viral replication in cells of the heterologous host has also already been reported in in vitro studies with sheep choroid plexus cells infected with a genotype B strain and goat fibroblasts infected with an A4 strain [14,33]. According to Chebloune et al., the restrictive replication of CAEV in sheep fibroblasts is due to the abnormal proteolysis of the *env* glycoprotein and the decreased synthesis of the *gag* protein. The cleavage of the precursor envelope glycoproteins may explain the persistence of the immune activity since disrupted proteins are still functional but also the continuous antibody production in heterologous infections, despite the presence of very low proviral loads [33]. In addition, the absence of viral replication in cell tissues of non-host species could have forced SRLV isolates to adapt and replicate more in macrophages of the lung and the mammary gland which could have contributed to the persistence of cross-species transmission events [12,34].

Similar observations were made in the lung and cerebellum, with detection of viral RNA upon homologous infection, albeit to a lower extent as in the mammary gland, while no viral RNA was detected upon heterologous infection. The low virus RNA load detected in the cerebellum was not associated with the presence of lesions, while this was difficult to judge in the lung since similar pathological lesions were also observed in the lung of control animals.

Since SRLV induced histopathological lesions and clinical disease are known to result from a degenerative inflammatory immune response, we compared cytokines’ mRNA expression levels at 9 months *p.i.* between homologous and heterologous infections. Overall, IFN-γ was the most upregulated cytokine in the mammary gland, cerebellum and lung of sheep and goats infected with their homologous strains. IFN-γ is mainly produced by NK cells and activated CD8+ T cells and has multiple functions. It acts as a mediator between the innate and adaptative immune response, and one of its main antiviral functions is to activate macrophages. In the context of SRLV infections, IFN-γ, TNF-α and GM-CSF have however been shown to activate the SRLV promoter and induce viral replication [35,36]. Thus, instead of blocking infection, it can lead to a cycle of continuous immune activation and to eventual immunological exhaustion and disease progression. The most consistent and elevated upregulation of IFN-γ mRNA was detected in the mammary gland after homologous infection of sheep and goats, and this was also the organ in which the highest viral loads and most severe histopathological lesions were found. Besides the upregulation of IFN-γ mRNA, a moderate increase in TNF-α, IL-10 and TGF-β mRNA was also observed in the mammary gland of genotype B infected goats. That correlates with the increase in the pro-inflammatory cytokine TNF-α previously reported in the udder of MVV-affected sheep [37], while IL-10 was found in the subscapular node of CAEV affected goats [37]. According to Murphy et al., the expression of TNF-α is responsible for the activation of the CAEV-promotor leading to an increase in viral replication [36].

In the lung, only the genotype A infected sheep with high viral load expressed an increased level of IFN-γ mRNA. Some other sheep expressed an increased level of IL-10 mRNA which is consistent with the study of Woodall et al. [38] who reported an association between an increase of these cytokines and the presence of lesions in the lung. The increased expression of IFN-γ in the mammary gland and lung correlated thus with the presence of SRLV and confirmed the importance of these tissues as sites of SRLV replication [32]. In contrast, the increased IFN-γ mRNA expression that was detected in the cerebellum did not correlate with detection of high viral loads or histopathological lesions. Hence, it seems that the effect of IFN-γ on SRLV replication differs depending on the target organ, and might thus be different on brain macrophages (microglia) compared to macrophages found in lung or mammary glands.

Although the ELISA results indicated no difference in seroconversion after homologous and heterologous infections, the AGID results whereby genotype B infected sheep remained seronegative stimulated us also to study the humoral immune response in more detail. We mapped the antibody response against the CA and SU (Env) proteins of the used inocula using a pepscan analysis. Both proteins are known to contain immunogenic epitopes and are commonly used in diagnostic tests [5,21,23,39]. The results confirmed that within the CA and SU proteins, most antibodies are produced against the known immunodominant region of the p25 capsid antigen [40] and the surface portion of the envelope SU5 region [29]. Importantly, the analysis showed that for both SRLV strains, more antibodies were produced against these epitopes after homologous compared to heterologous infections and that the response was higher and more consistent against the SU5 region compared to the immunodominant region of the capsid protein. The analysis of the antibody response over time against these immunodominant epitopes furthermore showed that the antibody response developed faster (between 1 and 3 months *p.i.*) in genotype B infected animals than in genotype A infected animals (between 3 and 9 months *p.i.*). Although hypothetical, this might be related to the fact that the inoculum dose of the genotype B strain was 100-fold higher than for the genotype A strain. So, the stronger antibody response against the SU protein after homologous compared to heterologous infections was inversely correlated with protection against virus replication in target organs and observed histopathological lesions. The phenomenon whereby virus-specific antibodies facilitate the viral entry into the host cells and increase viral infectivity and disease severity was first described in flaviviruses and is called antibody-dependent enhancement (ADE) of an infection [41]. A potential role for ADE in SRLV infection has already been suggested based on the results of vaccination studies. The vaccination of sheep with a viral vector containing a MVV sequence namely resulted in an increased viremia [42], and goats were reported to suffer from acute arthritis after they were vaccinated with an inactivated CAEV vaccine [43]. Others have argued against the occurrence of ADE in SLRV infections. CAEV-specific antibodies were namely shown to enhance virus entry in macrophages, but did not result in an increased production of virions [18], and it was suggested that SU5-specific antibodies probably do not bind intact virus particles since they are not neutralizing and therefore can play no role in ADE [44]. Although our results do not allow us to draw conclusions on this topic, they at least argue to study this in more detail and consider potential differences between monocytes/macrophages and epithelial cells of target organs and the role of others than SU5-specific antibodies.

Lastly, we also studied the efficacy of intra- and cross-species SRLV transmission under our experimental conditions. Despite the fact we tried to reproduce conditions that are also observed in intensive indoor farms (direct contact exposure, no fences or segregation between animals, shared equipment, communal feeder and watering), no horizontal transmission occurred between animals during the study period. This was somewhat unexpected since horizontal transmission to contact animals was reported before. Ilius et al. estimated the transmission rate of MVV to be about 1000 times faster in sheep housed indoors than sheep raised in prairies [45], and horizontal transmission was shown in other experimental studies to occur after 10 to 22 months in goat farms [46]. It thus seems that our study period of 9 months was too short to allow the transmission or the detection thereof. Moreover, another study previously reported the absence of horizontal SRLV transmission in non-dairy goats living in close cohabitation for 12 months [47]. Besides the duration of the study period, the efficient ventilation system operating in our experimental facilities could also have been a reason for the inefficient SRLV transmission between animals. To get additional insights in the reason for the observed absence of horizontal transmission, future studies should foresee in the collection of saliva, milk or urine to test for virus excretion over time.

## 4. Materials and Methods

### 4.1. Isolation, Virus Stock Production and Characterization of Genotypes A and B SRLV Strains

Virus stocks of the genotype A and B strains were produced via the successive cultivation of peripheral blood mononuclear cells (PBMCs) isolated from whole blood of naturally SRLV infected sheep and goat and splenic macrophages isolated from SRLV negative sheep, following a protocol adapted from Petursson et al. [48]. The genotype A and B strains were characterized as subtype A1 and B1 subtype strains, respectively, in a previous study [49]. After isolation of PBMCs over a Histopaque (Sigma-Aldrich, St. Louis, MO, USA) cell gradient, cells were washed twice in RPMI-1640 (Thermo Fisher Scientific, Waltham, MA, USA) and plated at a concentration of 10^6^ cells/mL in a 6-well plate. The culture medium consisted of RPMI-1640 supplemented with 1 mM of glutamine, 10 mM of HEPES, 0.1 mM of 2-mercapto ethanol, 1% of gentamicin, 2% of fungizone, 1000 IU/mL of penicillin and 20% of fetal calf serum, and cells were cultivated at 37 °C for 10 days. This specific medium adapted from Gorrell et al. [50] stimulated the differentiation of monocytes into macrophages and activated virus replication. Ten days later, 5 mL of genotypes A and B culture supernatant were collected and inoculated overnight on splenic macrophages isolated from SRLV negative sheep which were seeded into 80 cm^2^ culture flasks 24 h earlier. After one day, a new culture medium was added to the flask without removing the virus supernatant. The cell culture was incubated for 5 days before flasks were put in the −80 °C. After freeze/thawing, the supernatant was centrifuged and aliquoted, and the virus titre was determined. The viral titres of genotype A and B strains were determined using the method of Reed–Muench using splenic macrophages seeded in 24-well plates. In this way, genotype A and genotype B virus stocks were produced with a titre of 10^2.66^ TCID_50_/mL and 10^4.66^ TCID_50_/mL, respectively. Both virus stocks were stored at −80 °C before inoculation.

### 4.2. Experimental Infection and Sample Collection

A total of 21 Texel sheep and 20 Saanen goats aged between 1 and 2 years old, originating from SRLV-free herds, were used in this experimental study. To ensure their SRLV negative infection status, animals were tested in serology and with an in-house qPCR detecting genotype A and B strains [49]. At the beginning of the experiment, 6 sheep and 3 goats were intratracheally inoculated with 1 mL of the genotype A strain (dose: 10^2.66^ TCID_50_/mL) and 3 sheep and 5 goats with 1 mL of the genotype B strain (dose: 10^4.66^ TCID_50_/mL). The difference in inoculation dose between both strains means that we only compared results from genotype A infected sheep and goats, and genotype B infected sheep and goats. Five control animals (3 sheep and 2 goats) and 18 contact animals (9 sheep and 9 goats) were left uninfected. At 4 weeks post infection (*p.i.*), groups of 3 inoculated sheep or goats with the genotype A or B strain, respectively, were moved to separate pens, and 3 contact sheep or goats were added to evaluate potential intra- and cross-species transmissions. An overview of this experimental set-up can be found in Appendix A. Blood samples were collected from all animals every two weeks until 4 months *p.i.* and then every month until the end of the experiment. The experiment lasted for 9 months. Each time, sera and leucocyte pellets were immediately prepared and stored at −20 °C and −80 °C, respectively, until use. At the time of euthanasia, samples of 4 organs (lung, cerebellum, mammary gland and synovial membrane) from the SRLV inoculated animals were collected for downstream qPCR analysis and histopathology. The experimental protocol was approved by the joint ethical committee of CODA-CERVA and the Scientific Institute of Public Health Belgium (procedure agreement 20170529-01), and was conducted in BSL2 animal facilities (CODA-CERVA, Machelen, Belgium).

### 4.3. Histopathology

Histology was performed on lung, cerebellum and mammary gland samples. Immediately after sample collection, tissues were fixed in 4% formaldehyde for a minimum of 48 h. Formol fixed tissue samples were embedded in paraffin wax. Five µm sections were made and dried overnight at 40 °C. Sections were then stained with haematoxylin-eosin (HE), and the slides were examined and scored by a pathologist. For the lung, assigned scores were: 0 (no important histopathological lesions), 1 (limited interstitial and peribronchiolar lymphocytic infiltrates), 2 (moderate interstitial and peribronchiolar lymphocytic infiltrates with sometimes the presence of interstitial lymphocytic aggregates), 3 (multiple sites of interstitial and peribronchiolar lymphocytic aggregates). For the mammary gland, assigned scores were: 0 (normal glandular structures; no important histopathological lesions), 1 (limited amount of periacinar lymphocytic infiltrate and activated epithelium with the presence of intraluminal debris and macrophages), 2 (productive acini with focal atrophic and destructed acini, prominent lymphoblastic aggregates, capillary congestion and occasional interstitial proliferation of fibroblasts), 3 (massive lymphoblastic conglomerates, massive lymphocytic/blastic infiltration in the acini, atrophic and destructed acini and calcification in the luminae of several tubules) [14]. For the cerebellum, no histopathological lesions were observed, neither in the control nor in the infected animals, and they all received the score 0 (no important histopathological lesions).

### 4.4. Serological Methods

The presence of SRLV antibodies in collected sera was determined using two commercial AGID tests (AGID-CAEV p28 kit (Idexx, Westbrook, ME, USA) and Maeditect kit (Apha Scientific, Addlestone, Surrey, UK)) and a commercial ELISA (ID screen MVV/CAEV indirect (IDvet, Grabels, France)) according to the manufacturer’s instructions.

### 4.5. Determination of the CA and SU Gene Sequences of the Genotype A and B Strain

Different methods were used to obtain the sequence of the CA (0.8 kb) and SU (1.6 kb) proteins of the genotype A and B strains which were used as inocula. Sanger sequencing was used to obtain nucleotide sequences of the genotype B strain, while NGS using MinION (Oxford Nanopore Technologies (ONT), Oxford, UK) and Illumina technology (Illumina Inc., San Diego, CA, USA) was used to determine the sequence of the genotype A strain.

For the whole genome sequencing of genotype A using MinION, the viral RNA was extracted and amplified using in-house methodologies at PathoSense BV (Belgium). Amplified DNA was sequenced on a MinION R.9.4 Flow cell (ONT, Oxford, UK) using the Ligation Sequencing Kit LSK109 (ONT, Oxford, UK). Resulting raw data were acquired in MinKNOW (v4.1.22, ONT, Oxford, UK) and subsequently processed in an automated in-house bioinformatics pipeline. In short, fast5 files were basecalled using Guppy basecaller (-c dna_r9.4.1_450bps_hac.cfg; v.3.6, ONT, Oxford, UK), followed by demultiplexing, adapter trimming and quality filtering using qcat (v1.1.0, ONT, UK), and NanoFilt (v2.7.1, [51]), respectively. Filtered reads were then used to generate a de novo SRLV genome using Canu (v2.0; [52]). A final SRLV draft genome was polished using graphmap (v.0.5.2; [53]) and medaka (v1.0.0; ONT, Oxford, UK) after which target genes p25 (CA) and gp135 (SU) were extracted for downstream analyses. All software were run using default settings.

In parallel, the Illumina technology was also used to sequence the genotype A strain genome. In order to do that, RNA was extracted from cell cultured virus according to the NetoVIR protocol [54]. Briefly, cell lysate was filtered on a 0.8 µM membrane and submitted to heavy nuclease treatments prior to column-based nucleic acid extraction Nucleospin RNA virus kit (Macherey-Nagel, Düren, Germany). DNA was removed from the extract using Baseline-ZERO DNase (Lucigen, Madison, WI, USA), and the resulting RNA was purified and concentrated using a RNA Clean & Concentrator-5 kit (Zymo Research, Irvine, CA, USA). First strand cDNA was obtained from 10 µL RNA using random hexamers and SuperScript IV Reverse Transcriptase (Thermo Fisher Scientific, Waltham, MA, USA) according to the manufacturer’s instructions. Second-strand cDNA synthesis was performed using the NEBNext Ultra II nondirectional RNA second-strand synthesis module kit (New England Biolabs, Ipswich, MA, USA). The resulting cDNA was purified and concentrated using a DNA Clean & Concentrator-5 kit (Zymo Research, Irvine, CA, USA). The sequencing library was prepared using the Nextera XT DNA Library Preparation Kit (Illumina Inc., San Diego, CA, USA) with 1 ng of input DNA, according to the manufacturer’s instructions. Sequencing was performed on an Illumina MiSeq NGS sequencer at the Genomic Service Facility (VIB-UAntwerp Center for Molecular Neurology, Antwerp, Belgium) using a MiSeq reagent kit version 3—2 × 300 bp (Illumina Inc., San Diego, CA, USA) and yielding over 5.5 million paired-end reads. Ribomosomal RNA reads were filtered out using MIRA v5rc2 [55], and read quality trimming was performed with Trimmomatic v0.38 [56]. De novo assembly was achieved by combining two assemblers iva v1.0.9-2 and SPAdes v3.13.0 on a subset of 35,000 filtered reads [57,58].

For the genotype B strain, the CA region was amplified via 2 overlapping fragments obtained using the following primers: First fragment (582 nt) with P101 (5′-TGGGGATGAGACCTGAGACA-3′) and P103 (5′-GCTGATATTACCCATTGTAGGCA -3′) and the second fragment (684 nt) with P105 (5′-CTCAGGCAGGAGGAGCATT-3′) and P107 (5′-GTTGCGGGCCATTTCTCATT-3′). For the SU region, three overlapping fragments were amplified: The first fragment (483 nt) with P758 (5′-CCCAGGTAAGTAAGCAAATAGGG-3′) and P760 (5′-GGGTAAGGCACACAGAATCT-3′), the second fragment (674 nt) with P184 (5′-TCAGGGAAGTAAGCTAAGGTGT-3′) and P186 (5′-TGGTCCCCATTATCCCTGTTATA-3′) and the third fragment (682 nt) with P168 (5′-GGAGAATGTGTGCAGAGACC -3′) and P 170 (5′-CTCCCAGAGAAGCCCCTG -3′). All primers used to characterize the CA and the SU regions of genotype B strain were designed by using Primer3 (v4.1.0) based on the genome sequences of different B1 SRLV strains present in the NCBI Genebank database [59].

The PCR reactions for P105-107 (CA) and P168-170 (SU) were conducted as follows: activation of the FastStart Taq DNA polymerase (Roche, Basel, Switzerland) at 95 °C for 15 min, 45 cycles of denaturation at 94 °C for 30 s, annealing at 54 °C for 1 min, and extension at 72 °C for 1 min. For other amplifications with P101-103 (CA), P758-760 (SU) and P184-186 (SU), the cycling conditions were the same except for the annealing which was performed at 52 °C. The final reaction volume consisted of 25 µL of 2×Fastart PCR Master, 2 µL of each primer (0.4 µM), 16µL of RNase-free water and 5 µL of DNA. Amplified products were separated and visualized by electrophoresis on a 0.8% agarose gel containing 0.5% GelRed Nucleic Acid Gel staining (Biotium, Fremont, CA, USA). Specific bands were purified using the QIAquick Gel extraction Kit (Qiagen, Hilden, Germany) following the manufacturer’s instructions. Fragments were sequenced on an ABI 3130xl (Applied Biosystems, Carlsbad, CA, USA), and sequence analysis was conducted with BioEdit (version 7.2.5) software [60].

All obtained sequences via NGS and sanger sequencing were aligned with reference genotype A and B strains obtained from the NCBI Genebank (NCBI, Rockville Pike, Bethesda, MD, USA) database using ClustalW included within the software MEGA (v7.0) [61]. Nucleotides sequences were translated into amino acids with MEGA (v7.0) and verified within the Protein Basic Local Alignment Search Tool (BlastP) online tool (https://blast.ncbi.nlm.nih.gov/Blast.cgi?PAGE=Proteins accessed on 4 August 2021) for the correct open-reading frame. All obtained sequences were submitted to the NCBI GeneBank (NCBI, Rockville Pike, Bethesda, MD, USA) database under the following accession numbers (genotype A strain full sequence: MZ484405; CA and SU sequences of genotypes B MZ351192, MZ351193).

### 4.6. Pepscan Analysis

A pepscan analysis was performed to study the development of the antibody response against epitopes of the CA and SU proteins of genotype A and B strains. A set of overlapping peptides (15 aa) with an offset of 4 aa and an overlap of 11 aa were synthetized and purified based on the complete sequence of the CA and the SU proteins of the genotype A and B strains used to inoculate the animals. For the genotype A strain, the analysis consisted of 50 peptides for the CA and 180 peptides for the SU, while for the genotype B strain, a total of 51 peptides for the CA and 134 peptides for the SU proteins were included in the pepscan testing. The peptide synthesis was performed by JPT Peptide Technologies using the SPOT-synthesis technology (JPT Peptide Technologies GmbH, Berlin, Germany), and each peptide was biotinylated (Biotides) at the N-terminus to allow their selective immobilization in 96-well streptavidin-coated plates (Thermo Fisher Scientific, Waltham, MA, USA). Wells were coated with 0.2 nM of Biotides resuspended in PBS with 0.05% Tween 20 and 40% of DMSO, then washed and blocked for 30 min at room temperature with a solution of PBS, 400 µM of Biotin (Sigma-Aldrich, St. Louis, MO, USA) and 20% of sucrose (Sigma-Aldrich, St. Louis, MO, USA). Newly coated plates were sealed and stored overnight at 4 °C until use. Sera from (i) a negative control sheep and goat, (ii) 3 sheep and 3 goats inoculated with the genotype A strain and (iii) 3 sheep and 3 goats inoculated with the genotype B strain collected at 1, 3, 6 and 9 months *p.i.* were tested in the pepscan. Sera were diluted 1/100 in a blocking buffer and incubated for 1 h at room temperature on a shaking incubator at 300 rpm. After incubation of the coated peptides with the serum samples, wells were washed 4 times using a washing solution composed of PBS with 0.05% Tween. Afterwards, wells were incubated with a solution of anti-ruminant IgG-HRP conjugate antibodies (IDvet, Grabels, France) for 30 min at room temperature on a shaking incubator at 300 rpm. After the second washing step, plates were revealed with a substrate solution (TMB) for 15 min and stopped with a stop solution (IDvet, Grabels, France). The optical density was subsequently read at 450 nm (OD450) with an ELISA reader.

The reactivity of each serum for each peptide was calculated by dividing the OD450 value obtained for the test sample over the OD450 value for that peptide of a negative control serum. The signal of a certain peptide was considered to be positive if its OD450 sample/negative value was more than 2 times the mean OD450 sample/negative over all peptides within the same protein and the same serum [62].

### 4.7. Sample Preparation, DNA and RNA Isolation, cDNA Production

Leucocytes pellets were prepared by adding 8.5 mL of hemolysis buffer (16.6 g NH_4_Cl, 2.0 g NaHCO_3_, 0.185 g EDTA disodium salt per L H_2_O; pH 7.4) to 1.5 mL of EDTA blood [63], and tubes were incubated for 20 min at room temperature. Samples were then centrifuged for 10 min at 3000× *g*, and the pellet was resuspended in 200 µL of phosphate buffer saline (PBS). DNA extraction was performed using the QIAmp DNA Mini kit (Qiagen, Hilden, Germany), following manufacturer’s instructions.

For organs, 0.5 cm^3^ of lung and cerebellum were homogenized in 1 mL of phosphate buffered saline (PBS) with 2 zirconium beads by high-speed shaking (4 min, 25 Hz) in a TissueLyser II (Qiagen, Hilden, Germany). The same was conducted for the mammary gland and the synovial membrane, replacing zirconium beads by two 5 mm stainless steel beads to improve the homogenization. After homogenization, tubes were centrifuged at 10,000 rpm for 1 min, and the supernatant was used for RNA extraction. Total RNA was extracted using the RNeasy Mini Kit (Qiagen, Hilden, Germany) according to manufacturer’s instruction. Extracted RNA was then treated with Turbo DNase (Thermo Fisher Scientific, Waltham, MA, USA) to remove contaminating genomic DNA and eluted in 40 μL RNAse-free water following manufacturer’s instructions. A total of 4 µL of DNase treated RNA were subsequently retrotranscribed to cDNA using the M-MLV reverse transcriptase system (Thermo Fisher Scientific, Waltham, MA, USA). For each reaction, a mix of 4 μL 5 × first strand buffer, 2 μL of DTT (final conc: 10 mM), 1 μL of dNTP mix (final conc: 0.5 mM), 0.2 μL of 10 × hexanucleotide mix, 0.5 μL M-MLV RT (final conc: 100 U), 8.3 μL H2O and 4 µL RNA was prepared and incubated at 37 °C for 45 min. This was followed by an inactivation step at 95 °C for 10 min.

### 4.8. SRLV Detection by qPCR

The presence of genotype A and B SRLV strains in blood and organs was determined using previously published qPCRs targeting the gag gene region of genotype A and B strains [49,64]. In sheep, the sensitivity and specificity of the qPCR was reported to be 88% and 100% respectively. In goats, the sensitivity and specificity of the qPCR was reported to be 83.3% and 100% respectively [49]. β-actin detection was included as an extraction control. For each qPCR reaction, a master mix consisting of 4 µL RNase free water, 10 µL 2× FastStart TaqMan Probe Master kit (Roche, Basel, Switzerland) and 1 µL of a pre-mix containing the forward primer (final conc: 500 nM), reverse primer (final conc: 500 nM) and probe (final conc: 200 nM) was prepared. A total of 5 µL of DNA extract or cDNA were added per reaction, and the following amplification program was used: 10 min at 95 °C, followed by 45 cycles of 15 s at 95 °C and 45 s at 60 °C. Plates were ran on a LightCycler 480 Real-Time PCR system (Roche, Basel, Switzerland). Samples with Ct values between 40 and 45, which were associated with a characteristic amplification curve, were considered to be positive and are mentioned as Ct > 40 [49].

A standard curve was generated by testing a 10-fold serial dilution series of the genotype A and B viral stocks in the qPCR described above, and used to convert the Ct values obtained for samples to TCID_50_/_mL_ or TCID_50_/g.

### 4.9. Preamplification

To obtain sufficient cDNA to study cytokine gene expression in target organs, a preamplification was performed using the TaqMan PreAmp master mix (Thermo Fisher Scientific, Waltham, MA, USA) following manufacturer’s instructions. A mix of primers including those targeting the selected reference genes and the cytokines of interest was prepared at a final concentration of 0.1µM per primer (Table 4). The reaction mix consisted of 10µL of TaqMan PreAmp master mix, 5µL of pooled assay mix and 5µL of cDNA. The preamplification program consisted of a denaturation step at 95 °C for 10 min followed by 14 cycles of amplification for 5 s at 95 °C and 4 min at 60 °C. The preamplification uniformity of cDNA of reference genes and selected cytokine genes in the lung, cerebellum and mammary gland was calculated following manufacturer’s protocol. Normalized Ct values of the different target genes to the reference genes before (cDNA) and after preamplification (PreAmp) were calculated according to the following formulas: ΔCt [cDNA] = Ct [cDNA target gene] − Ct [cDNA reference gene] and ΔCt [PreAmp] = Ct [PreAmp target gene] − Ct [PreAmp reference gene], respectively. The ΔΔCt was subsequently determined from the difference of the two ΔCts: ΔΔCt = ΔCt [PreAmp] − ΔCt [cDNA]. A PreAmp uniformity is considered ideal when ΔΔCts are close to zero and PreAmp values are within the range −1.5 and +1.5.

### 4.10. Cytokines Gene Expression by qPCR and Relative Quantification

Previously described primers by Sahu et al. (Table 4) [65] were used for the detection of reference genes (ACTB, HMBS, YWHAZ, HPRT1) in different organs of sheep and goats using SYBR Green I technology. Each qPCR reaction consisted of a mix of 5 µL preamplified cDNA, 10 µL of 2 × LightCycler 480 SYBR Green I Master (Roche, Basel, Switzerland), 1 μL per primer (final concentration of 0.5 µM) and 3 µL of H_2_O that was subjected to the following cycling conditions: 95 °C for 5 min to activate the polymerase, followed by 45 cycles of denaturation at 95 °C for 10 s and annealing/extension at 60 °C for 10 s and 72 °C for 10 s. At the end of the cycle, the melting curve was determined to control the specificity of the reaction products. All samples were run in duplicate on a LightCycler 480 real-time PCR system (Roche, Basel, Switzerland).

Primers and probes for the detection of following selected cytokines (IFN-α, IFN-β, IFN-γ, TNF-α, IL-8, IL-10, TGFβ1, GM-CSF) (Table 4) were designed using Primer-BLAST software (http://www.ncbi.nlm.nih.gov/tools/primer-blast accessed on 4 August 2021) and published ovine and caprine genome sequences obtained from NCBI Genebank database (NCBI, Rockville Pike, Bethesda, MD, USA). In order to use the same qPCR on both species, specificity of primers and probes was verified in BLAST (https://blast.ncbi.nlm.nih.gov/Blast.cgi accessed on 4 August 2021). The FastStart TaqMan Probe Master kit (Roche, Basel, Switzerland) was used for qPCR reactions. Each qPCR reaction included a master mix of 4 µL RNase free water, 10 µL of 2× FastStart probe master buffer, 1 µL of a pre-mix containing the forward primer (10 µM), reverse primer (10 µM) and probe (4 µM) and 5 µL of preamplified cDNA. The following qPCR program was used: polymerase activation and denaturation at 95 °C for 10 min, followed by 45 cycles of 15 s at 95 °C and 45 s at 60 °C. All samples were run in duplicate on a LightCycler 480 real-time PCR system (Roche, Basel, Switzerland).

All primers and probes were manufactured by integrated DNA technologies (IDT, Coralville, IA, USA). A list of the sequences used is shown in Table 4.

Prior to the analysis of the samples collected during this study, it was determined for each target organ which reference genes were appropriate for normalization via the GNorm method [66]. ACTB, HMBS and YWHAZ were selected for the lung. HMBS and HPRT were selected for the cerebellum, and HPRT and YWHAZ were selected as the most suitable reference genes for mammary gland. Those references genes were used in each target organ of both species. Target Ct values and reference gene Ct values were used for the quantification and normalization of the qPCR results in qBasePlus (v3.1; Biogazelle, Zwijnaarde, Belgium) software taking technical and experimental errors into account [67,68]. Cytokine expression levels of the infected animals were expressed relative to the average of the control group. All bar charts were prepared using GraphPad Prism (v9; GraphPad Software, San Diego, CA, USA).

### 4.11. Statistical Analysis

All statistical analyses were performed using GraphPad Prism (v9; GraphPad Software, San Diego, CA, USA). The Mann–Whitney test was used to evaluate whether differences in time to seroconversion were observed between sheep and goats infected with the genotype A and B strains, respectively. The total number of positive organs found in sheep and goats after a genotype A or genotype B infection, respectively, was compared using the Fischer’s exact test. The Kruskal–Wallis test was used to examine whether differences in proviral loads were present in different organs after infection of sheep and goats with genotype A and genotype B strains, respectively.

## 5. Conclusions

This large-scale experimental infection study confirmed that genotypes A and B strains are able to infect both sheep and goats, but that viral replication in target organs and the induction of histopathological lesions are more prominent after homologous (genotype A in sheep, genotype B in goats) compared to heterologous (genotype A in goats, genotype B in sheep) infections. The viral load and lesions observed in the mammary gland and lung at 9 months post homologous infection correlated with an increased IFN-γ mRNA expression, supporting the adverse effect this antiviral cytokine could have on SRLV replication and induced disease. In contrast, the increased IFN-γ expression in the cerebellum seems to contribute to the limitation of virus replication. Although no difference in the time to seroconversion was observed after homologous and heterologous genotypes A and B strain infections in those animals in which a productive infection was started, the pepscan revealed important differences in host-specific antibody responses against capsid and envelop proteins after genotype A and B strain infections. The antibody response against the SU protein was inversely correlated with protection against virus replication in target organs and observed histopathological lesions, suggesting the need for an in-depth evaluation of a potential role of ADE in SRLV infection. Finally, the intra- and cross-species transmissions of SRLV strains between small ruminants via horizontal transmission proved to be inefficient under our experimental conditions.

## Figures and Tables

**Figure 1 ijms-22-09824-f001:**
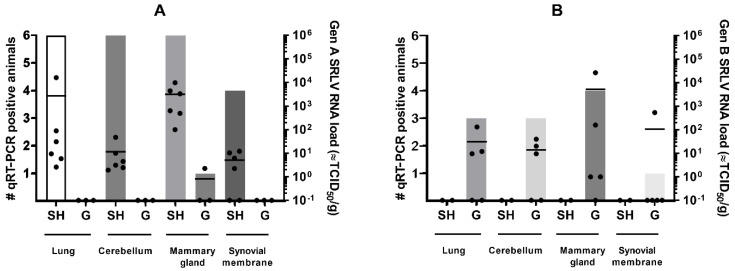
SRLV RNA detection in target organs of sheep (SH) and goats (G) at 9 months after homologous and heterologous genotype A (panel **A**) and B (panel **B**) strain infection. Bars indicate the number of qPCR positive animals. Each dot corresponds to one animal and represents the viral load. The mean viral load is indicated by the horizontal line.

**Figure 2 ijms-22-09824-f002:**
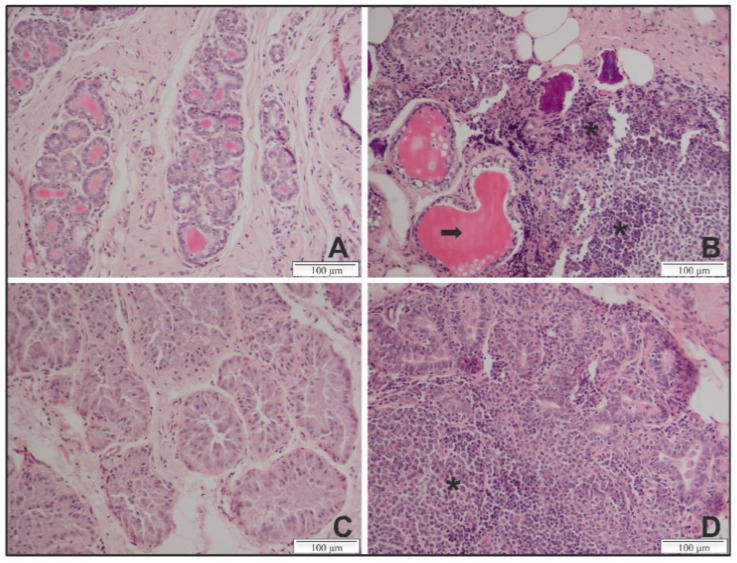
Representative histopathological lesions in the mammary gland of genotype A and B SRLV strain infected sheep and goats at 9 months *p.i*. (**A**) Section from an uninfected sheep exhibiting structured acini and no lymphocyte infiltration. (**B**) Section from a sheep infected with a genotype A strain exhibiting a massive lymphocyte infiltration (asterisks) and a destruction of the acini structure (arrow). (**C**) Section from an uninfected goat exhibiting tissues with normal glandular structures and no lymphocytic infiltrate. (**D**) Section from a goat infected with a genotype B strain exhibiting lesions with massive lymphocyte infiltrations and an atrophy of the acini. Magnification ×20.

**Figure 3 ijms-22-09824-f003:**
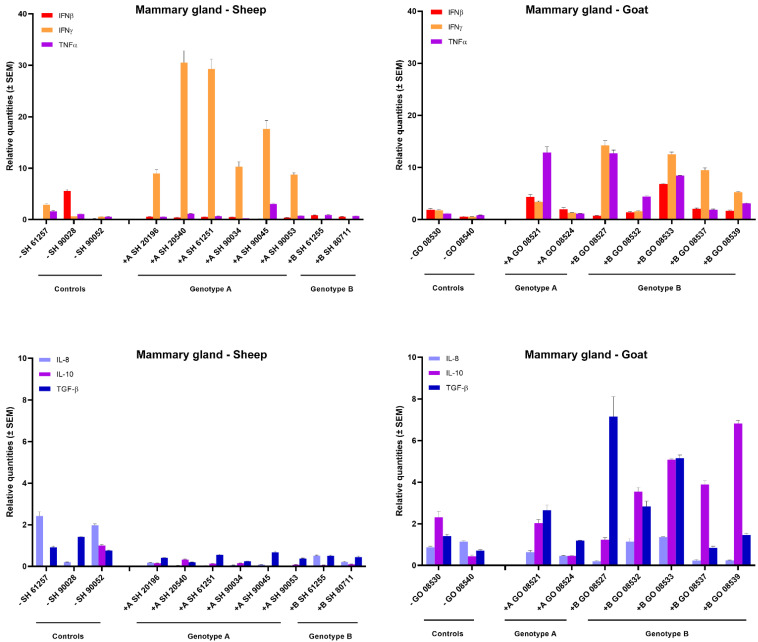
Cytokines’ mRNA expression in the mammary gland of genotype A and genotype B infected sheep and goats. mRNA expression was detected by qPCR for IFN-α, IFN-β, IFN-γ, TNF-α, IL-8, IL-10, TGF-β1 and GM-CSF.

**Figure 4 ijms-22-09824-f004:**
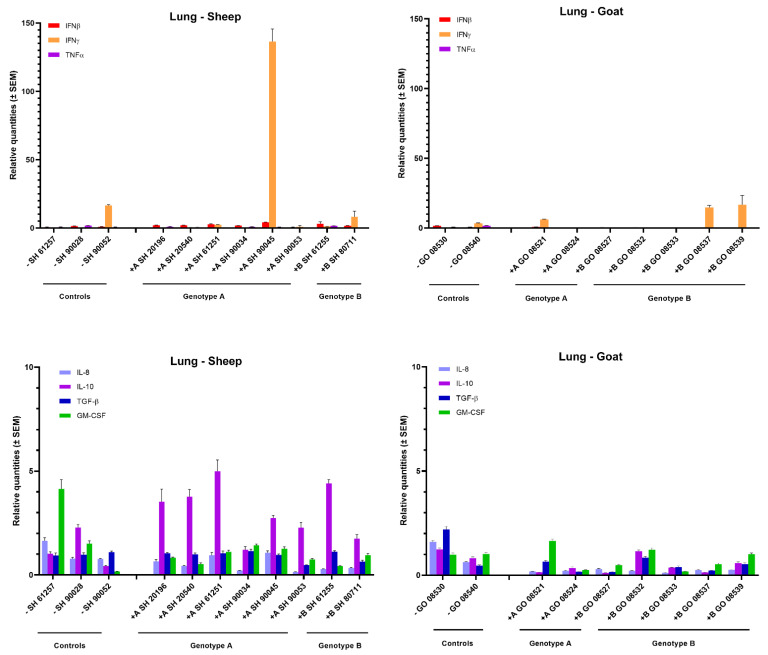
Cytokines’ mRNA expression in the lung of genotype A and genotype B infected sheep and goats. mRNA expression was detected by qPCR for IFN-α, IFN-β, IFN-γ, TNF-α, IL-8, IL-10, TGF-β1 and GM-CSF.

**Figure 5 ijms-22-09824-f005:**
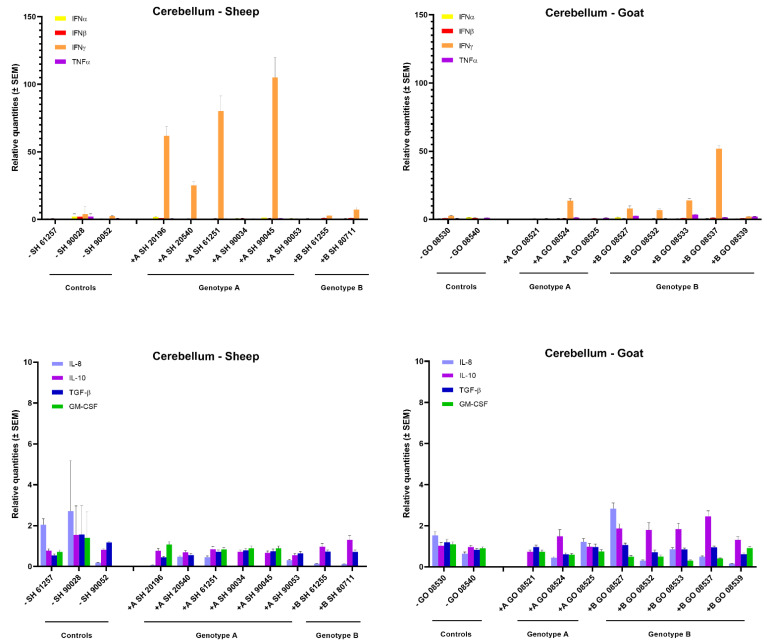
Cytokines’ mRNA expression in the cerebellum of genotype A and genotype B infected sheep and goats. mRNA expression was detected by qPCR for IFN-α, IFN-β, IFN-γ, TNF-α, IL-8, IL-10, TGF-β1 and GM-CSF.

**Figure 6 ijms-22-09824-f006:**
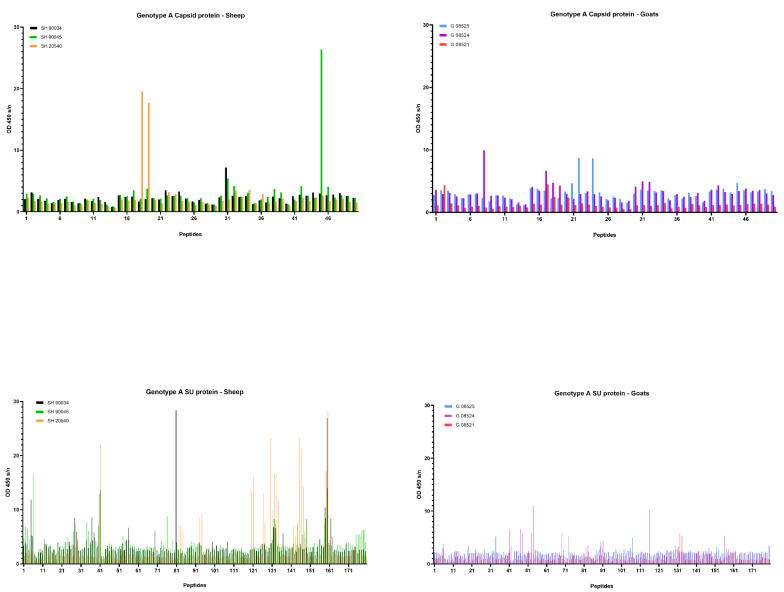
Pepscan analysis of sera from genotype A infected sheep and goats against linear peptides of the capsid and surface proteins of the genotype A strain used as inoculum. Sera were tested at a dilution of 1/100, and each bar represents the reactivity of one serum against one single peptide.

**Figure 7 ijms-22-09824-f007:**
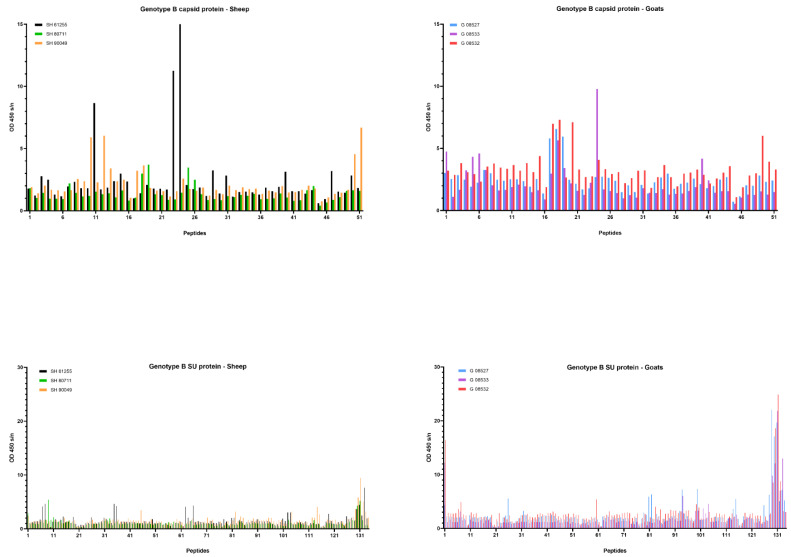
Pepscan analysis of sera from genotype B infected sheep and goats against linear peptides of the capsid and surface proteins of the genotype B strain used as inoculum. Sera were tested at a dilution of 1/100, and each bar represents the reactivity of one serum against one single peptide.

**Figure 8 ijms-22-09824-f008:**
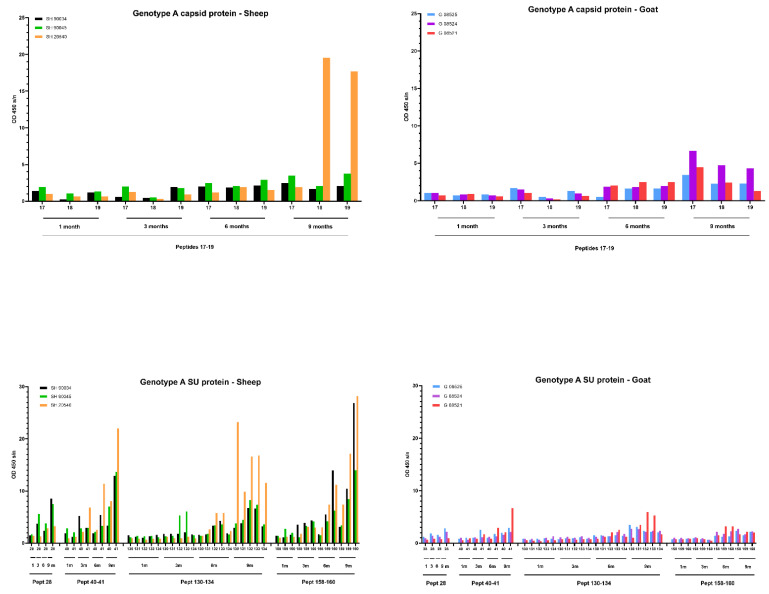
Detection of specific antibodies over time in sera collected from genotype A infected sheep and goats against selected epitopes of the capsid and surface proteins via pepscan analysis. Sera collected at 1, 3, 6 and 9 months (m) *p.i.* were tested at a dilution of 1/100, and each bar represents the reactivity of one serum against one single peptide.

**Figure 9 ijms-22-09824-f009:**
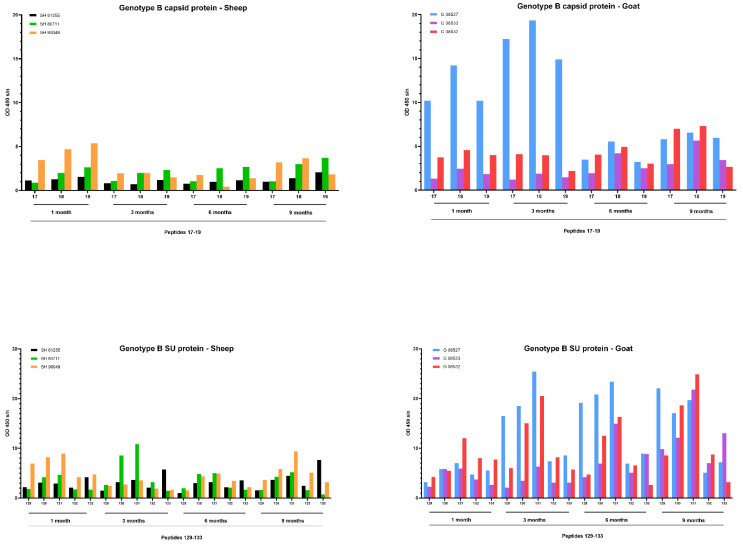
Detection of specific antibodies over time in sera collected from genotype B infected sheep and goats against selected epitopes of the capsid and surface proteins via pepscan analysis. Sera collected at 1, 3, 6 and 9 months *p.i.* were tested at a dilution of 1/100, and each bar represents the reactivity of one serum against one single peptide.

**Table 1 ijms-22-09824-t001:** First time point of provirus and virus-specific antibody detection in genotype A and B infected sheep and goats, and scores of histopathological lesions at the moment of euthanasia.

Inoculum	Sheep/Goats Number	Time to Become qPCR Positive (Weeks *p.i.*)	Time to Become ELISA Positive (Weeks *p.i.*)	Time to Become AGID Positive (Weeks *p.i.*)	Score of Histological Lesions in Lung	Score of Histological Lesions in Cerebellum	Score of Histological Lesions in Mammary Gland
Genotype A 10^2.66^ TCID_50_/animal	Sheep 90034	9	16	12	3	0	2
	Sheep 20196	24	28	16	3	0	2
	Sheep 61251	4	6	10	2	0	2
	Sheep 90053	4	16	16	3	0	3
	Sheep 90045	2	6	9	2	0	2
	Sheep 20540	6	6	9	2	0	3
	Goat 08525	Undetected	Undetected	Undetected	0	0	N.A
	Goat 08524	Undetected	Undetected	Undetected	1	0	1
	Goat 08521	4	9	16	1	0	1
Genotype B 10^4.66^ TCID_50_/animal	Sheep 61255	5	11	Undetected	2	0	1
	Sheep 80711	4	4	Undetected	2	0	1
	Sheep 90049 ^a^	4	8	Undetected	N.A	N.A	N.A
	Goat 08527	8	4	8	0	0	3
	Goat 08533	5	5	15	2	0	1
	Goat 08537	4	8	11	2	0	2
	Goat 08532	2	2	8	2	0	2
	Goat 08539	5	4	35	2	0	1
	Goat 08531 ^b^	N.A	N.A	N.A	N.A	N.A	N.A

0 = no important histopathological lesions; N.A = not available; ^a^ = sheep 90049 died unexpectedly at 32 weeks *p.i.*, and no organs were collected; ^b^ = Goat 08531 died at 2 weeks *p.i.* after a fight, and no samples were collected.

**Table 2 ijms-22-09824-t002:** Alignment of the immunodominant region of the capsid protein of different genotype A and B reference strains and the genotype A and B strains used as inocula in the experimental infection.

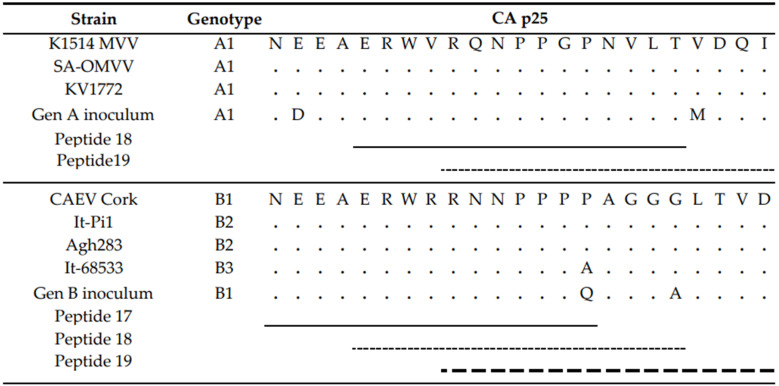

**Table 3 ijms-22-09824-t003:** Alignment of the immunodominant SU5 region of different genotype A and B reference strains and the genotype A and B strains used as inocula in the experimental infection.

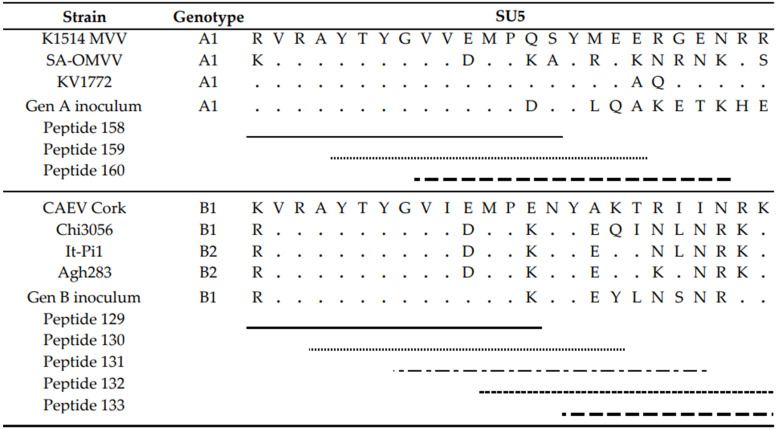

**Table 4 ijms-22-09824-t004:** Primers and probes used for the detection of selected reference genes and cytokines.

Gene	Description	Primer/Probe	Sequence (5′-3′)	Amplicon Size	Efficiency %	Reference or Source
*Reference genes*						
ACTB	β-actin	ACTB_F	CTCTTCCAGCCTTCCTTCCT	101	109	[65]
		ACTB_R	TAAAGGTCCTTGCGGATGTC			
HMBS	Hydroxymethyl-bilane synthase	HMBS_F	CTTGCCAGAGAAGAGTGTGG	115	104	[65]
		HMBS_R	CAGCCGTGTGTTGAGGTTTC			
YWHAZ	Tyrosine 3-monooxygenase activation protein zeta polypeptide	YWHAZ_F	TTCTGAGGTGGCTTCTGGAG	117	105	[65]
		YWHAZ_R	AGTCGAATGGGGTGTGTAGG			
HPRT	Hypoxanthine phosphoribosyltransferase 1	HPRT_F	CACTGGGAAGACAATGCAGA	102	102	[65]
		HPRT_R	ACACTTCGAGGGGTCCTTTT			
*Cytokines*						
IFN-α	Interferon alpha	IFN-α_F	TCGACAACTGAGGAGGGTCT	161	105	Primer-BLAST
		IFN-α_R	CCTCTGTGCTGAAGAGCTGG			
		IFN-α_Probe	TGGGTGGCAGCCAGTTGCAG			
IFN-β	Interferon beta	IFN-β_F	GATGAAGCAAGCACAGCAGT	181	107	Primer-BLAST
		IFN-β_R	TGGCTCCAGACGATTCATCT			
		IFN-β_Probe	CACCAGAGACTTCTCCAGCA			
IFN-γ	Interferon gamma	IFN-γ_F	CCATAACACAGGAGCTACCGAT	91	110	Primer-BLAST
		IFN-γ_R	GAGCAGTAAAGCTAAGAAGGAGC			
		IFN-γ_Probe	ACTACTCCGGCCTAACTCTCTCCT			
TNFα	Tumor necrosis factor alpha	TNF-α_F	ACGAACCCATCTACCAGGGA	77	95	Primer-BLAST
		TNF-α_R	TTCCGGCAGGTTGATCTCAG			
		TNF-α_Probe	AGCTGGAGAAGGGAGATCGCCT			
IL-8	Interleukine-8	IL-8_F	CATTCCACACCTTTCCACCC	124	108	Primer-BLAST
		IL-8_R	GGGGTCTAAGCACACCTCTTT			
		IL-8_Probe	TGAGAGTGGGCCACACTGCG			
IL-10	Interleukine-10	IL-10_F	ACCTGTGTTTAAGCTGTTTCCA	249	109	Primer-BLAST
		IL-10_R	AGAGGGCAGTCAGGGAAAAC			
		IL-10_Probe	TGCAGTTTCCATTCCAAGCC			
TGF-β1	Transforming growth factor beta 1	TGF-β1_F	CCTGCTGAGGCTCAAGTTAAAAGT	82	102	Primer-BLAST
		TGF-β1_R	CTGAGGTAGCGCCAGGAATT			
		TGF-β1_Probe	TGCTATATTTCTGGTACAGCTCCACGTGCTG			
GM-CSF	Granulocyte macrophage colony stimulating factor	GM-CSF_F	ACTGTGGTCTGCAGCTTCTC	109	101	Primer-BLAST
		GM-CSF_R	TGCTGTCGTTCAGAAGGCTC			
		GM-CSF_Probe	CACTCGCCAACCCAGCCCTG			

## Data Availability

All important data is included in the manuscript.

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
