# Peer review of "Species-Specific Humoral Immune Responses in Sheep and Goats upon Small Ruminant Lentivirus Infections Inversely Correlate with Protection against Virus Replication and Pathological Lesions"

_ijms, 2021, doi:10.3390/ijms22189824_

Round 1
Reviewer 1 Report
SRLV infection is a serious problem in sheep and goat around the world, and the measures to prevent this infectious disease have been tried to establish. In this study, the authors analyzed the immune response after infection of SRLV genotype A and B to goat and sheep, including epitope mapping and semi-quantification of mRNA of immune-related factors to reveal pathogenicity of SRLV infection. In Pepscan analysis, antibody response to immunedominant regions of the CA and SU on homologous infection was stronger than on heterologous infection, and it was correlated with histopathological lesions.
After 9 months post-inoculation, virus genome sequence might be changed but not determined. The virus genome changes might affect to humoral immune response, especially for antibody reaction against virus epitopes. The authors should determine at least CA and SU region of virus or provirus at 9 months.
To evaluate intra and cross species horizontal transmission at section 2.3, the it should be necessary to evaluate whether there was any discharge of the virus from saliva, milk or urine. The information on the viral genome level in saliva, milk or urine should be added.
In the strength of pathogenicity and other evaluations, genotype A and B were compared. However, inoculation titer of genotype B was almost 100-times higher than genotype B. This difference might affect to the results. So, comparison in strength between genotype A and B should be deleted. And why different virus titer was used?
At line 127, It is better to describe clearly the target sample type, provirus (DNA) or virus genomes (RNA), such as "qPCR analyses for provirus" in the first sentence.
At line 138, provirus genome was detected intermittently on some animals. This result depended on the sensitivity of the qPCR assay. So, the authors should add the information on the limit of detection of this assay in materials and methods section, even if that was already described in previous studies. It was important information to understand it.
At line 140-142, the authors mentioned that SRLV infection was detected earlier by qPCR than ELISA. The authors should add this result was obtained under this experimental condition, because this inoculation titer might be higher than natural infection. It might be lead to misunderstand.
In Table 1, the timing of AGID positive was earlier than qPCR positive in Sheep20196. Normally, antibody against pathogens increased after the increase of antigen level like other animals. Why this animal showed different reaction?
In histopathological analysis on figure 2, it is better to perform IHC analysis to understand the distribution of virus.
In pepscan analysis, why did only 3 of all, not all, animals infected with SRLVs were selected?
In Table 2 and 3, please indicate the area of each peptide by underlining, etc.
At line 461, IFN-gamma is known to activate the SRLV promoter and IFN-gamma mRNA was upregulated in cerebellum. However, proviral genome could not be detected in that organs. The authors should discuss on it in discussion section.
At line 478, IFN-gamma is one of important factors for virus infection control and not specific marker of SRLV presence. This sentence might lead to misunderstand so the authors should revise this section.
Author Response
SRLV infection is a serious problem in sheep and goat around the world, and the measures to prevent this infectious disease have been tried to establish. In this study, the authors analyzed the immune response after infection of SRLV genotype A and B to goat and sheep, including epitope mapping and semi-quantification of mRNA of immune-related factors to reveal pathogenicity of SRLV infection. In Pepscan analysis, antibody response to immunedominant regions of the CA and SU on homologous infection was stronger than on heterologous infection, and it was correlated with histopathological lesions.
After 9 months post-inoculation, virus genome sequence might be changed but not determined. The virus genome changes might affect to humoral immune response, especially for antibody reaction against virus epitopes. The authors should determine at least CA and SU region of virus or provirus at 9 months.
First of all we would like to thank the reviewer for reading our manuscript and making all these interesting remarks.
We agree that SRLV mutate during their replication in a host. It can however be hypothesized that this evolution would be different in each individual animal and it is was not in the scope of our study (neither in time, budget nor research questions) to study this evolution over time in detail. Although we can agree that viral mutations can influence the humoral immune response, our results show that the humoral immune reactions in all sheep and all goats is rather similar, especially for the most important immune-dominant epitopes. Together with the fact that most animals seroconverted rather soon after infection and that SRLV spend most of their time in the latent status, we estimate that potential mutations would only have had a minor impact on the detected antibody response and don’t influence the outcome of our study.
To evaluate intra and cross species horizontal transmission at section 2.3, the it should be necessary to evaluate whether there was any discharge of the virus from saliva, milk or urine. The information on the viral genome level in saliva, milk or urine should be added.
We acknowledge that it would have been interesting to collect saliva, milk or urine and check it for virus presence over time. This would have provided additional insights as to why we haven’t observed horizontal transmission in this study. Unfortunately, such samples were not collected and can thus not be analyzed. We added this as an important suggestion for future studies in our manuscript (line 540-542).
In the strength of pathogenicity and other evaluations, genotype A and B were compared. However, inoculation titer of genotype B was almost 100-times higher than genotype B. This difference might affect to the results. So, comparison in strength between genotype A and B should be deleted. And why different virus titer was used?
As the reviewer will know, production of virus stocks of SRLV is very cumbersome and takes a lot of time. In preparation of our experiment, we produced a genotype A stock with a titer of 102.66 TCID50/ml and a genotype B stock of 104.66 TCID50/ml. Since we had no time and budget to preliminary check the minimal infectious dose of these viruses, we decided to use them as such.
In the description of the methods and the analysis of the results, we were very careful to compare only genotype A infections between sheep and goats, and genotype B infections between sheep and goats, and thus not to compare genotype A and B infections in sheep and genotype A and B infections in goats. This was explicitly mentioned at line 577 and 578.
The only time we compared something between genotype A and genotype B infection in the discussion (line 414-417), it was clearly mentioned that different inoculation doses were used.
At line 127, It is better to describe clearly the target sample type, provirus (DNA) or virus genomes (RNA), such as "qPCR analyses for provirus" in the first sentence.
The sentence on line 127 was modified as suggested by the reviewer (now line 128) and this was also clarified at line 160.
At line 138, provirus genome was detected intermittently on some animals. This result depended on the sensitivity of the qPCR assay. So, the authors should add the information on the limit of detection of this assay in materials and methods section, even if that was already described in previous studies. It was important information to understand it.
The sensitivity and specificity of our in-house qPCR was added in the M&M in lines 750-752 .
At line 140-142, the authors mentioned that SRLV infection was detected earlier by qPCR than ELISA. The authors should add this result was obtained under this experimental condition, because this inoculation titer might be higher than natural infection. It might be lead to misunderstand.
We thank the reviewer for this interesting remark. We added this information in the manuscript in line 141.
In Table 1, the timing of AGID positive was earlier than qPCR positive in Sheep 20196. Normally, antibody against pathogens increased after the increase of antigen level like other animals. Why this animal showed different reaction?
We have no clear indication why it took longer before proof of an SRLV infection was found in this specific animal. The most plausible explanation is that less target cells became infected upon inoculation compared to other animals, making that the amount of proviral DNA remained for a long time under the limit of detection of the qPCR test. The virus that was present at undetectable levels in qPCR was however sufficient to elicit an antibody response that was detected by AGID and ELISA, but also somewhat later than in most other animals.
In histopathological analysis on figure 2, it is better to perform IHC analysis to understand the distribution of virus.
In this part of our already extensive study, we focused on the induced pathological lesions in target organs and therefore performed HE stainings. We agree that IHC stainings could have been interesting, but we did not know whether viral proteins would be detectable at the moment that samples were collected. It was therefore decided to study virus presence in the target organs via qPCR analysis.
In pepscan analysis, why did only 3 of all, not all, animals infected with SRLVs were selected?
This decision was made based on budgetary grounds. A lot of expensive reagents were necessary to screen the selected sera for all the peptides. We think testing a selection of 3 genotype A infected sheep and goats and 3 genotype B infected sheep and goats led to clear and meaningful results (for heterologous infections, a maximum of 3 animals were available).
In Table 2 and 3, please indicate the area of each peptide by underlining, etc.
The question of the reviewer is not clear to us. The amino acids shown in table 2 make-up the immunodominant peptide region of the capsid protein and the amino acids shown in table 3 form the immunodominant region of the surface protein called SU5. These tables show the homology (and heterogeneity) between different reference MAE and CAE strains in these regions, including the virus strains that were used as inoculum in our study.
At line 461, IFN-gamma is known to activate the SRLV promoter and IFN-gamma mRNA was upregulated in cerebellum. However, proviral genome could not be detected in that organs. The authors should discuss on it in discussion section.
This observation was discussed in lines 484-486. It indeed seems that IFN-gamma has a different effect on SRLV replication depending on the organ (promoting SRLV replication in mammary gland and lungs, while suppressing it in the cerebellum). It might be that brain macrophages react differently than macrophages in other tissues. This is something that deserves further study.
At line 478, IFN-gamma is one of important factors for virus infection control and not specific marker of SRLV presence. This sentence might lead to misunderstand so the authors should revise this section.
We thank the reviewer for pointing out this misunderstanding. We agree that IFN-gamma should not be seen as a specific marker for the presence of SRLV. The sentence was therefore modified (now lines 485-486).
Reviewer 2 Report
The authors present a very interesting and comprehensive study of small ruminant lentivirus infections of sheep and goats. This in vivo study sought to address the differences in infection of Maedi-Visna virus (MVV) and Caprine arthritis encephalitis virus (CAEV) in what might arguable be considered the natural hosts, sheep and goats, respectively. The study also examined the impacts of CAEV infection in sheep and MVV infection in goats. The dynamics of these infections are subsequently determined over a nine month period, including virological, immunological and pathological impacts on the animals. The potential for transmission to in contact animals was also evaluated, but not detected. The study is very well described, and the results support the Authors’ conclusions. The manuscript will be of interest to those working on these interesting viruses.
I have made some suggestions below for the authors to consider:
Line 14 As the abstract is separate from the main text the abbreviations should be explained in full.
Line 22 suggest replacing “what” with “which”
Line 25 suggest deleting “what”
Line 26 suggest revision “Thus requires an in depth”
Line 28 suggest revision “contact animals was detected.”
Line 55 suggest revision “Despite their close genetic relationships”
I do not think “proximity” is the appropriate word in this context.
Line 138 suggest revision “while others intermittently tested positive”
Line 148 suggest revision “contact animals that had been housed with the inoculated animals for 8 months”
Line 150 suggest revision “contact animals was detected during the entire experiment (data not shown).”
Line 178 suggest revision “No statistical differences were detected in the viral load between the different organs.”
Line 188 suggest revision “detection in target organs of sheep (SH) and goats (G)”
Line 189 The individual panels on the figure should be labelled rather than being described as “left” and “right”.
Line 216 – The sizes of the scale bars are illegible.
Line 222 suggest revision “Magnification ×20”
Line 269 suggest replacing “moment” with “timing”
Line 273 suggest using a more specific description than “some weeks later” by quoting actually range.
Line 410 The superscripts of 10,000 to 100,000 require “superscripting”.
Line 412 suggest replacing “gets started” with “is initiated”
Line 418 suggest revision “ELISA would be recommended since antibodies remained detectable”
Line 646-655 The oligonucleotide sequences in this paragraph could be added to Table 4 and the table title modified to something more generic. For example, "Oligonucleotides used in this study."
Line 729 Reaction components should be quoted as final concentrations where appropriate (e.g. DTT, dNTP, units of M-MLV RT). See line 784.
Author Response
The authors present a very interesting and comprehensive study of small ruminant lentivirus infections of sheep and goats. This in vivo study sought to address the differences in infection of Maedi-Visna virus (MVV) and Caprine arthritis encephalitis virus (CAEV) in what might arguable be considered the natural hosts, sheep and goats, respectively. The study also examined the impacts of CAEV infection in sheep and MVV infection in goats. The dynamics of these infections are subsequently determined over a nine month period, including virological, immunological and pathological impacts on the animals. The potential for transmission to in contact animals was also evaluated, but not detected. The study is very well described, and the results support the Authors’ conclusions. The manuscript will be of interest to those working on these interesting viruses.
We would like to thank the reviewer for taking the time to read our manuscript, the positive evaluation and for all the interesting comments he/she made. Responses to all questions and suggestions are provided below (and in the manuscript).
Line 14. As the abstract is separate from the main text the abbreviations should be explained in full
Abbreviations were replaced by full words in the abstract (line 1).
Line 22 suggest replacing “what” with “which”
This modification was made on line 22
Line 25 suggest deleting “what”
The word was deleted as suggested and the sentence was modified on line 25
Line 26 suggest revision “Thus requires an in depth”
The sentence on line 26 was modified as suggested by the reviewer (now line 27).
Line 28 suggest revision “contact animals was detected.
The sentence on line 28 was modified as suggested by the reviewer (now line 29).
Line 55 suggest revision “Despite their close genetic relationships”
The sentence on line 55 was modified according to the reviewer’s suggestion. We agree that the word “relationship” is more adequate in this context (now line 56).
Line 138 suggest revision “while others intermittently tested positive”
The sentence on line 138 was modified as suggested by the reviewer (now line 139)
Line 148 suggest revision “contact animals that had been housed with the inoculated animals for 8 months”
The sentence on line 148 was modified as suggested by the reviewer (now line 150)
Line 150 suggest revision “contact animals was detected during the entire experiment (data not shown).”
The sentence on line 150 was modified as suggested by the reviewer (now line 152)
Line 178 suggest revision “No statistical differences were detected in the viral load between the different organs.”
The sentence on line 178 was modified as suggested by the reviewer (now line 181)
Line 188 suggest revision “detection in target organs of sheep (SH) and goats (G)”
The sentence on line 188 was modified as suggested by the reviewer (now line 192)
Line 189 The individual panels on the figure should be labelled rather than being described as “left” and “right”.
Labels A and B were added above each panel and these modifications were added in the description (now line 193)
Line 216 – The sizes of the scale bars are illegible.
The size of the scale bar was increased for clarity.
Line 222 suggest revision “Magnification ×20”
Change was made (now line 227).
Line 269 suggest replacing “moment” with “timing”
The sentence on line 269 was modified as suggested by the reviewer (now line 274)
Line 273 suggest using a more specific description than “some weeks later” by quoting actually range.
A range was added to the manuscript on line 274 and “data not shown” was replaced by “Table 1”on line 278.
Line 410 The superscripts of 10,000 to 100,000 require “superscripting”.
The sentence on line 410 was modified as suggested by the reviewer (now line 416)
Line 412 suggest replacing “gets started” with “is initiated”
The sentence on line 412 was modified as suggested by the reviewer (now line 418)
Line 418 suggest revision “ELISA would be recommended since antibodies remained detectable”
The sentence on line 418 was modified as suggested by the reviewer (now line 424)
Line 646-655 The oligonucleotide sequences in this paragraph could be added to Table 4 and the table title modified to something more generic. For example, "Oligonucleotides used in this study."
We see the point of the reviewer but we preferred to separate the M&M of the sequencing and the cytokines gene expression. If the reviewer agrees, we would like to keep the primers of the sequencing analysis in the text to avoid a mix-up.
Line 729 Reaction components should be quoted as final concentrations where appropriate (e.g. DTT, dNTP, units of M-MLV RT). See line 784.
As requested, we added the final concentrations where appropriate (lines 739-740 and 752-753).
Reviewer 3 Report
This manuscript describes a careful, systematic study of the course of infection of small ruminant lentiviruses and the ensuing immune responses in homologous (MVV in sheep, CAEV in goats) and heterologous (the reverse) hosts. The authors report productive infection in all four settings, but higher virus loads in the homologous hosts. Interestingly, these homologous settings also yield higher interferon-gamma and antiviral antibody titers. In light of this correlation between viral titers and antiviral responses, the authors raise the possibility of antibody-dependent enhancement (ADE) of infection in this system.
It is surprising to read (line 506) the reference to ADE in HIV as if this were an established phenomenon. This is supported by citing a review (ref. 42) from 2003, but even this review states that ADE is “controversial” in the case of HIV. I would urge the authors to be more cautious in this area: I do not believe that ADE has been demonstrated or widely accepted in HIV.
Author Response
This manuscript describes a careful, systematic study of the course of infection of small ruminant lentiviruses and the ensuing immune responses in homologous (MVV in sheep, CAEV in goats) and heterologous (the reverse) hosts. The authors report productive infection in all four settings, but higher virus loads in the homologous hosts. Interestingly, these homologous settings also yield higher interferon-gamma and antiviral antibody titers. In light of this correlation between viral titers and antiviral responses, the authors raise the possibility of antibody-dependent enhancement (ADE) of infection in this system.
It is surprising to read (line 506) the reference to ADE in HIV as if this were an established phenomenon. This is supported by citing a review (ref. 42) from 2003, but even this review states that ADE is “controversial” in the case of HIV. I would urge the authors to be more cautious in this area: I do not believe that ADE has been demonstrated or widely accepted in HIV.
We see the point of the reviewer regarding the limited evidence of ADE in the very well-studied HIV. We therefore decided to remove this aspect from the manuscript (line 512). All other parts on the discussion on ADE and SRLV remain very relevant and lead to the suggestion to study this in more detail.
Round 2
Reviewer 1 Report
I think the revised manuscript is worthy of publication in this journal to help researchers better understand the pathogenesis of SRLV infection after additional minor revisions.
On AGID result of Sheep 20196 in Table 1, the authors mentioned some suggestion in the response to reviewers’ comment. I think it should be better to add it to discussion section.
Each immunodominant region in Table 2 and 3 includes several epitopes for Pepscan analysis, I think. So, please show each epitope area used for Pepscan analysis with underline in each Table.
Author Response
We warmly thank the reviewer for her/his positive comments. All necessary changes were made in the manuscript.
Point-by-point response to the comments of the reviewer:
Reviewer 1:
I think the revised manuscript is worthy of publication in this journal to help researchers better understand the pathogenesis of SRLV infection after additional minor revisions.
We warmly thank the reviewer for her/his positive comments. All necessary changes were made in the manuscript.
On AGID result of Sheep 20196 in Table 1, the authors mentioned some suggestion in the response to reviewers’ comment. I think it should be better to add it to discussion section.
The suggestions that were mentioned previously in the rebuttal letter were added in the manuscript on lines 432-438.
Each immunodominant region in Table 2 and 3 includes several epitopes for Pepscan analysis, I think. So, please show each epitope area used for Pepscan analysis with underline in each Table.
Thanks for these clarifications. Tables 2 and 3 were modified and we now underlined the AA constituting the different peptides used in the Pepscan analysis covering these immunodominant domains. Tables were modified in Excel to make the lines perfectly aligned to the AA sequences and to make it easier for the editor to modify it afterwards. We don’t know why the dots line in peptide 19 is similar to peptide 18 while in the Excel file, every dots lines are different. The original Excel file will be sent to the editor for further modifications.